# Non-native plant removal and high rainfall years promote post-fire recovery of *Artemisia californica* in southern California sage scrub

**Diane M. Thomson**[1]*, **Wallace M. Meyer, III**[2], **Isobel F. Whitcomb**[1]

**1** W.M. Keck Science Department, Claremont, California, United States of America, **2** Department of Biology, Pomona College, Claremont, California, United States of America

* dthomson@kecksci.claremont.edu

**Data Availability Statement:** All data are available from the Dryad database (https://doi.org/10.5061/dryad.hdr7sqvjb).

## Abstract

Non-native plant invasions, changes in fire regime, and increasing drought stress all pose important threats to biodiverse mediterranean-climate shrublands. These factors can also interact, with fire and drought potentially creating opportunities for non-native species to establish dominance before native shrubs recover. We carried out post-fire demographic monitoring of the common native shrub *Artemisia californica* in a southern California sage scrub fragment for 7 years, including several with very low rainfall. Experimental removals of non-native plants were included for the first 4 years. We quantified *A. californica* post-fire crown resprouting and seedling emergence, and tested effects of precipitation, non-native plants, and their interactions on seedling and adult survival. Only 7 *A. californica* were confirmed as resprouts; almost all individuals established after the fire from seedlings, with 90% of emergence occurring in the second growing year after fire (spring 2015). Higher spring precipitation increased both adult and seedling survival. Non-native grasses and forbs rapidly recolonized control plots, but the removal treatment reduced non-native cover by nearly 60%. For seedlings, non-native removal reduced the probability of dropping leaves by start of summer drought and increased survival both directly and through positive interactions with rainfall. Non-native removal also reduced mortality in smaller adult plants. By 2020, mean *A. californica* canopy area was nearly four times greater in non-native removal plots. These findings reinforce the high vulnerability of sage scrub habitat to post-fire loss of shrub cover and potential type conversion, particularly with increasing drought frequency and in stands and species with limited crown resprouting. Yet they also illustrate the potential for targeted management of non-natives immediately after fire to promote recovery of native shrubs in this increasingly endangered community.

## Introduction

Mediterranean biome regions are important global biodiversity hotspots, harboring high species richness and under intense pressure from human impacts [1]. Mediterranean-climate ecosystems have undergone extensive losses to urban and agricultural development, and negative

**Funding:** The author(s) received no specific funding for this work.

effects of fragmentation threaten the habitat that remains [2]. Growth in the wildlands-human interface facilitates the spread of non-native invasive species that can reduce native plant richness [3]. Many mediterranean-climate communities evolved with and depend upon fire, but whether fires promote persistence of native diversity depends on the frequency and intensity [4]. Anthropogenic changes in fire regime pose another major threat to these habitats [2, 5, 6].

Interactions between fire and non-native plant invasion can create positive feedback loops that further promote loss of native cover and diversity [7, 8]. Fires open up opportunities for non-natives to invade, particularly annual grasses and forbs that respond to disturbance more rapidly than native perennials [9]. Non-native grass and forb invasion in turn magnify fine fuel loads, potentially increasing ignition risk, fire frequency, or fire intensity [10, 11]. These changes in fire regime can lead to habitat degradation and even type conversion, meaning a major and persistent shift in community structure such as replacement of native shrubland by non-native grassland [12].

Southern California supports two native shrubland communities heavily impacted by ongoing type conversion, chaparral and sage scrub [13]. By 1994, sage scrub had declined to an average of 36% shrub cover compared to the 60–90% observed 60 years earlier [14]. Similarly, between 1953 and 2016 nearly 30% of study plots in chaparral shifted from shrub to herbaceous dominance of cover [15]. Annual grasses such as *Bromus diandru*s, *Bromus madritensis*, and *Avena fatua* are the most abundant non-native species in these type-converted communities, along with annual forbs such as *Brassica* spp. and *Hirschfeldia* spp. Increased fire frequency is associated with higher non-native grass and forb cover in both chaparral and sage scrub [14, 16–18]. Yet the directions of any causal relationships remain unclear; these correlations could reflect the effects of fire disturbance on non-native plants, the effects of non-native plants on ignition risk and fuel loads, or both.

The first few years after fire in southern California shrublands may be a critical window determining whether native communities recover or non-native invasion takes hold [19]. Once annual grasses are established they reduce native seedling germination and survival, through suppressive effects of litter [20, 21] and alteration of soil water availability [22] or soil microbial communities [23]. In contrast, high shrub cover helps control non-native grasses and forbs through shading, nitrogen depletion, and herbivory [24–27]. As a result, rapid shrub recovery after fire is critical to prevent non-native grasses from establishing dominance [12, 28]. Reintroducing shrubs to restore type-converted, non-native grasslands has proved very difficult [5, 29].

This key role of the years immediately after fire potentially also amplifies effects of increased drought and climate warming on native shrub communities. Low rainfall years can reduce shrub survival and recruitment [24, 30], and more arid regions have proved at greater risk of non-native invasion after fire [12, 19]. Between 2011 and 2018, southern California experienced an intense drought that included the driest conditions of the last 1200 years [31]. This drought followed a period of longer-term drying trends over the last several decades, increasing the urgency of understanding how precipitation affects post-fire recovery and non-native invasion [32].

Both fire ecology and non-native invasions in many respects have been extensively investigated in southern California shrublands. Yet post-fire monitoring [33] and experimental tests of non-native plant effects [21] generally are not integrated into the same studies. Moreover, most previous southern California post-fire research followed large-scale burns in 1978 and 1993, before the onset of recent intense drought conditions (but see [34]). While a few studies document the importance of water availability to native shrubs in sage scrub [35], none have yet developed statistical models that link rainfall with shrub demographic rates.

We tracked recovery of the common shrub *Artemisia californica* over the first 7 years after a small-scale fire in a sage scrub fragment, combined with experimental removal of non-native plants. We quantified *A. californica* post-fire resprouting and seedling emergence, as well as survival and growth of both seedlings and established plants. Experimental removals of non-native grasses and forbs were carried out during the first four years to evaluate their effects on post-fire demography and recovery of *A. californica*. We tested effects of precipitation, non-native plants, and their interactions on seedling and adult survival.

## Materials and methods

### Study system

As is typical in other mediterranean climate regions, southern California shrublands include an evergreen, sclerophyllous community type (chaparral) and a drought-deciduous, softer-leaved one (sage scrub) [36]. Sage scrub concentrates along the Pacific coast from San Francisco, CA (latitude 37.3) to El Rosario, Mexico (latitude 30.06), with this distribution in some areas extending eastward towards the Mojave Desert [37]. Sage scrub supports high total plant diversity, but local within-patch species richness tends to be low [36]. Species composition varies widely over short distances, and several subtype classifications have been proposed based on specific plant associations [38]. Estimates of historic habitat loss vary between 40% and 90%, with much of the remaining range considered degraded [5, 39]. Sage scrub supports more than 60 plant and more than 30 animal taxa classified as rare, threatened, or endangered [40].

Historic fire return intervals in sage scrub are uncertain but have been estimated at about 30 years [41]. In some areas with higher non-native grass cover, fire intervals have shortened to less than 8 years [14]. Post-fire recovery in both sage scrub and chaparral is mostly driven by species present at the time of fire, which regenerate either through resprouting or germination from the seed bank [42]. The majority of woody shrubs are facultative seeders that can both resprout and regenerate from seed, with the balance of these two processes varying across species and habitats [42]. Chaparral species resprout at higher rates after fire than sage scrub dominants, but may experience little to no recruitment without fire [41]. In contrast, sage scrub shrubs can germinate in gaps without fire disturbance [24]. Several studies suggest that the ratio of resprouting to recruitment from seed in sage scrub declines across a moisture gradient, from more mesic coastal to drier inland habitat [33, 43].

*Artemisia californica* is a suffrutescent sub-shrub that occurs in chaparral but is more common in sage scrub. Among the most widely distributed sage scrub species, *A. californica* is co-dominant at many sites [38, 39]. This species is both facultatively drought-deciduous and seasonally dimorphic, producing smaller leaves during summer [37]. *Artemisia californica* appears as a community dominant primarily in south coastal areas, replaced by species such as *Encelia farinosa* in interior regions [39]. Across sage scrub sites, *A. californica* reaches peak cover with intermediate temperatures and low litter [38]. *Artemisia californica* has been classified as a facultative seeder and can crown resprout after fire, but at lower rates than some other common co-occurring species such as *Salvia apiana* and *Eriodictyon trichocalyx* var. *trichocalyx* [37, 42, 44].

We conducted this study at the Robert J. Bernard Field Station, which supports fragments of intact sage scrub on 34 hectares embedded in a suburban landscape (34.8 ha; 34˚6' N, 117˚42' W; 348 m elevation; Claremont, CA, U.S.A.). Stands of sage scrub are bordered by roads and a matrix of anthropogenically altered habitats, including type-converted annual grassland. The exact time of last previous fire is unknown but extends back at least 60 years. The climate is mediterranean, with cool, wet winters and warm, dry summers. Winter rains typically start

in October and end in April or May with onset of summer drought. Growing year rainfall (September to August) averages 415.5 mm per year (n = 95 years with complete records; 1896–1978 NOAA Claremont Station; 2000–2020 Western Regional Climate Center data for Claremont, CA). Over the study period, growing year rainfall varied from 55.4% below to 79.3% above this mean; the first growing season after fire (2013–2014) experienced almost exactly mean precipitation (418 mm), while four of the six subsequent years were at least 24% below average (S1 Table).

Late in the 2013 dry season (September), an accidental fire burned 6.9 hectares containing two patches of sage scrub separated by a road (Fig 1). The western burned patch included approximately 1000 m$^2$ of sage scrub bordered by non-native grassland on two sides. The eastern patch encompassed about 4000 m$^2$ of sage scrub bordered by non-native grassland on all sides. *Artemisia californica* dominated pre-fire vegetation in both patches (mean ± one standard error 2012–2013 foliar cover, west: 47.4 ± 4.8%, east: 30.1 ± 3.7%), with *Eriodictyon trichocalyx* var. *trichocalyx* (west: 7.6 ± 1.2%, east: 16.1 ± 2.8%), and *Eriogonum fasciculatum* var. *foliolosum* (west: 5.9 ± 1.6%, east: 4.1 ± 1.5%) the next most abundant native shrubs. Both areas also supported a high cover of non-native grasses in the understory (west: 27.6 ± 4.4%, east: 53.3 ± 5.3%). *Bromus diandrus* and *B. madritensis* L. subsp. *rubens* were equally represented in the west patch, and *B. diandrus* about twice as abundant as *B. madritensis* in the east patch. Non-native forbs constituted less than 5% of pre-fire cover in both patches, with bare ground common (22.4%).

## Experiment and data collection

We established 12 plots spanning the west to east axis of the burned sage scrub area, each 10 m by 10 m. Plots were separated by a 5 m wide buffer. Eight plots were in the western and four plots in the eastern patch (Fig 1). Treatments were paired to control for the west to east pre-fire gradient in shrub and non-native grass cover. We randomly assigned one plot to each treatment within neighboring pairs, starting with the westmost boundary.

For the first four years after fire (spring 2014–2017), non-native grasses and forbs were hand weeded from plots assigned to the removal treatment. Removal began each year in mid-January and continued until collection of cover data started in the last week of March. Once per week during that time period, a team of two to three volunteers with experience in basic identification of local weedy species spent 20 minutes per treated plot removing non-natives. All non-native species were included, except that *Erodium* spp. proved difficult to control and were therefore targeted less.

We measured vegetative cover in all plots during years when treatments were applied (2014–2017), using point-intercept sampling on a grid. Nine transects were established from west to east in each plot, spaced at 1 m intervals from south to north. Along each transect, we sampled points at 0.5 m intervals (N = 162 per plot) between the last week in March and the first week in May. The identities of all species touching a straight edge held up and down from the soil surface were recorded for each sample point (foliar cover); as a result, total cover values can exceed 100%.

Demographic data for *A. californica* were collected from 2014 to 2020 at an annual census between June 14 and June 30, early in summer drought. In the first census after fire (2014), we individually tagged and mapped the locations of all *A. californica* in 10 plots. For one control and one removal plot adjacent to each other (6 and 7), high numbers of plants precluded tagging all individuals. All plants in the western halves of these two plots were tagged in June 2014, and any surviving plants in the eastern halves in November 2014. Plants were treated as distinct individuals if their stem bases were separate where entering the soil. At every annual

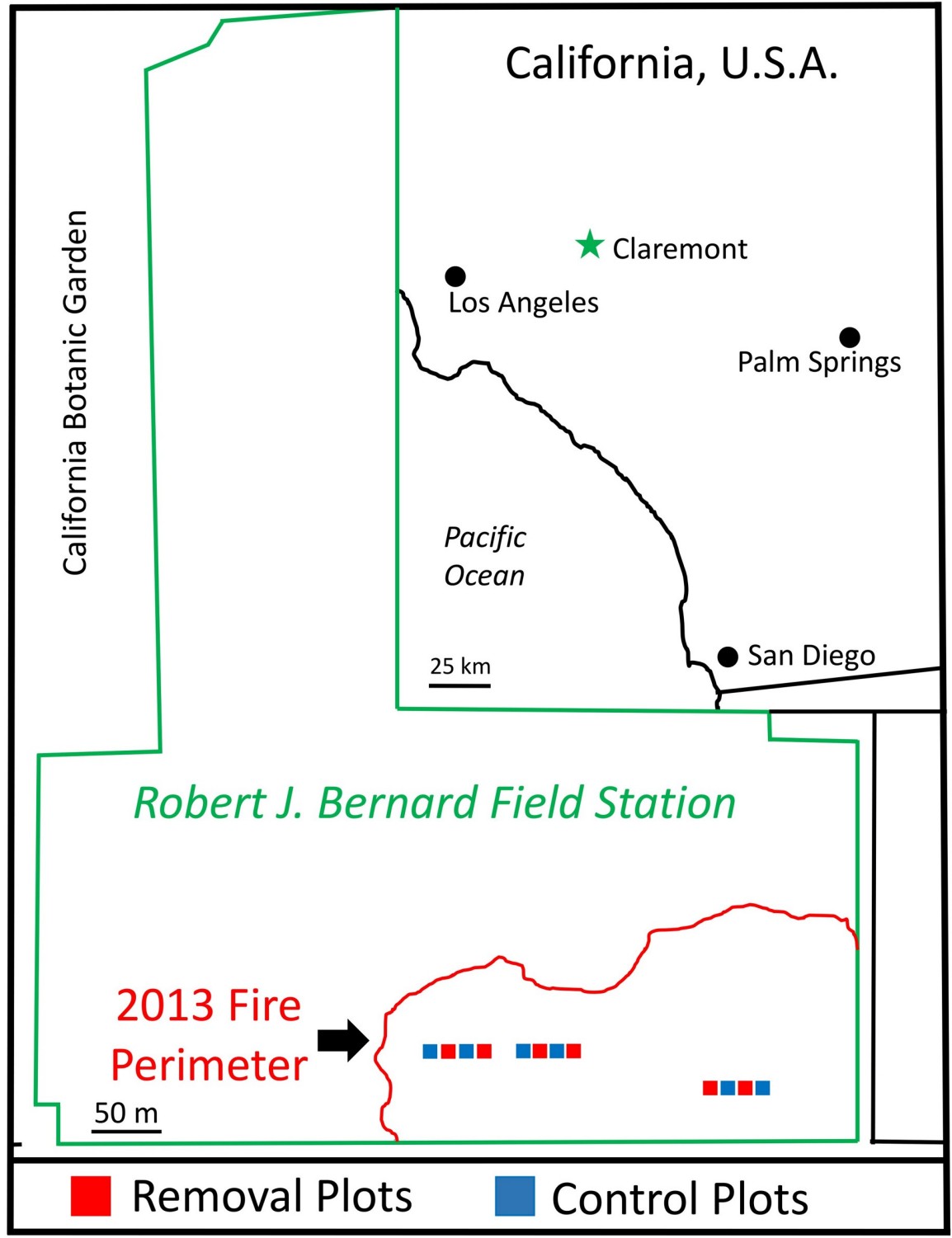

**Fig 1. Map of the study area at the Robert J. Bernard Field Station in Claremont, CA.** The filled squares represent the 12 study plots (100 m$^2$ each, red for non-native removal treatment and blue for control).

census, leaf condition of each plant was classified as either good, if leaves were still present and appeared to be photosynthetically active, or deciduous, if the plant had begun to shed leaves. We quantified size for all individuals as canopy volume, by recording plant height, plant diameter on the longest axis, and a second diameter perpendicular to the first.

All seedlings also were assigned individual tag numbers, mapped, and measured in most years after 2014 (2016–2018, 2020). In 2019, we counted all seedlings and mapped, tagged, and measured up to 6 per plot. Subsampling was used to assess seedling density and sizes in 2015, because of high emergence rates. We searched for seedlings in 2015 using a grid of points spaced at 2 m intervals within each plot ($N = 16$). At each sample point, seedlings within a 0.5 m radius were counted, scored for condition, and marked with a twist tie loop help in place by a plastic fruit fork. The seedling closest to the sampling point was measured. If no seedlings were found within 0.5 m, we repeated the same protocol at a 1 m radius. We then systematically searched any plot where no seedlings were detected at the sampling points. If fewer than five seedlings were measured from the sampling grid, additional sizes were recorded until reaching at least $N = 5$ per plot, or until all seedlings were measured. All seedling density data were collected in the same June census used for adult plants.

In 2015, we also marked seedlings in a subset of plots earlier in the spring to assess survivorship from germination to the June census. The two westernmost plots (one control, one removal) were surveyed between February 20 and March 1, and the next pair of plots between March 12 and April 3. A second control plot was added to the early April tagging, to increase sample sizes for control seedlings.

In some cases, seedlings were missed and appeared as untagged individuals in the following year. We used size records from known seedlings and first year plants to create criteria for assigning untagged individuals to a recruitment cohort. Untagged plants that were either less than 20 cm in height or smaller than 3000 cm$^3$ in canopy volume were assumed to be seedlings. This classification system correctly predicted 98.4% of records for confirmed seedlings ($N = 128$), and 97.0% of those for confirmed one-year old plants ($N = 34$).

*Artemisia californica* likely resprout from aboveground organs [36], but definitively identifying resprouting individuals in the first year after fire would have required excavating roots [41]. We classified plants tagged in June 2014 as likely seedlings unless they exceeded the size thresholds used in other years or could be confirmed as resprouts by the presence of dead stems. Keeley and Keely (1984) report May sizes for sage scrub shrubs that resprouted after fire well over our criteria (e.g., mean heights of 50–52 cm), although these values were combined across species.

## Data analysis

All statistical analyses were carried out in R (version 3.6.1). Treatment effects on non-native cover from 2014–2017 were quantified by first aggregating data for individual plots within each year. We determined the number of non-native species observed per sampling point, then calculated plot means for those values. A linear mixed effects regression in the package lme4 [45] was used to test for effects of treatment, time (year) since fire and their interaction, with plot as a random effect.

Records of *A. californica* emergence and survival included two years from the time period after removal treatments ended (2018–2020). We treated all plots in these two years as controls. To make sure this decision did not drive any of our findings, we also ran the adult and seedling survival analyses with data only from the time period when experimental manipulations were carried out (2014–2017). None of the qualitative results for effects of precipitation, non-native removal, and plant size change when data from 2018–2019 are excluded.

We aggregated seedling emergence data by plot and year, calculating seedling density as the total number of seedlings counted divided by the total area sampled. Transformation did not normalize these data, so effects of treatment and year were assessed with a linear model permutation test using the package lmPerm [46]. We compared the final distribution of *A. californica* plants in 2020 between plots that received control and removal treatments from 2014–2017, using a chi square goodness of fit test against the null hypothesis of a 50:50 ratio. Total *A. californica* canopy volume per plot in 2020 was then compared between removal and control treatments, using a t test after square root transformation.

We also quantified effects of non-native removal and precipitation on survival of both adults and seedlings, with binomial generalized linear mixed effects models in lme4. We used total precipitation from January through June as the rainfall predictor because this window captures the winter and spring months when *A. californica* is most visibly leafed out and growing (National Climate Data Center records, Claremont, CA). Spring precipitation values were normalized as a proportional deviation from the mean for 2000–2020. Both seedling and adult models included plot as a random effect to account for repeated measures. Adult survival models also included individual plant identity as a random effect and log transformed canopy volume as a fixed effect. We selected a final best fit model for both adults and seedlings by comparing models including different combinations of fixed effects and interactions with likelihood ratio tests (LRT) (S2 Table). Significance of fixed effects was assessed with bootstrapped LRT in the R package pbkrtest, comparing the best fit model selected by AIC with one that either added or dropped the individual parameter [47]. In total, analyses included $N = 1006$ records for marked seedling survival and N = 2785 annual transitions for 461 adult plants.

For the large spring 2015 cohort specifically, we compared June leaf condition between control and removal plants with a mixed binomial model including plot as a random effect. Estimates of early seeding survival (from March or April to June) drew on records from only one to two plots per treatment, so the frequency of survival was compared between control and removal with a Fisher's Exact Test (March: $N = 47$ control, $N = 26$ removal.; April: $N = 99$ control, $N = 47$ removal).

## Results

The removal treatment reduced cover of non-native grasses and forbs other than *Erodium* by nearly 60% relative to controls ($t = -3.44$, $df = 42.8$, $p = 0.001$), although with a marginal trend towards smaller effects over time ($t = 1.63$, $df = 34.0$, $p = 0.11$). *Erodium* cover did not differ between control and removal plots ($t = 0.26$, $df = 9.99$, $p = 0.80$), but increased after the first year until peaking at $15.3 \pm 4.8\%$ in 2016 (Table 1). Non-natives rapidly recolonized control plots, reaching mean foliar cover values of 49–60% by the third year after fire (Table 1). The Mediterranean annual grass *Bromus madritensis* dominated non-native cover, present at

**Table 1. Mean and standard error (in parentheses) values for total foliar cover of non-native grasses and forbs in control compared to removal plots over the four treatment years.**

| Year | Control (SE) | Removal (SE) | *Erodium* (SE) |
|------|-------------|-------------|---------------|
| 2014 | 18.0 (8.0) | 1.9 (0.8) | 3.2 (1.4) |
| 2015 | 49.4 (11.1) | 13.5 (3.6) | 8.4 (3.3) |
| 2016 | 60.2 (6.8) | 32.2 (5.9) | 15.3 (4.8) |
| 2017 | 48.7 (6.7) | 35.6 (7.1) | 9.5 (2.5) |

Cover for *Erodium* spp. is shown separately, averaged for both control and removal plots; removal treatments had no effect on *Erodium* cover.

17.4% to 42.3% of all sample points in control plots and 9.6% to 32.4% in removal plots from 2015–2017. *Bromus diandrus* also reached cover values over 10% in control plots during some years (S3 Table). The most common non-native forb after *Erodium* was *Brassica nigra*, observed at up to 11.4% of sampling points in control plots (2015).

Seven *A. californica* were confirmed as resprouts in June 2014, 3 in control and 4 in removal plots. The other 454 adult plants identified and tagged through 2019 recruited from seed (98.5% of total). Seedling emergence varied strongly spatially, but densities were far higher in the second growing year after fire (spring 2015) than any other year (Fig 2, $p < 0.0001$). In 2020, 54.3% of all adults were from the 2015 seedling cohort, compared to 19.1% from the 2019 cohort and 12.4% from the 2014 cohort. Non-native removal did not significantly change seedling emergence (Fig 2, $p = 0.2$).

Both higher spring rainfall and larger plant size improved adult survival (Table 2, Fig 3). Non-native removal significantly increased survival of the smallest adults, but this effect disappeared with increasing plant size (Table 2). Removal treatments also strongly improved seedling survival (Table 2, Fig 4; S1 Table). Higher spring rainfall both benefitted seedlings directly and enhanced the effects of non-native removal (Table 2). For the largest cohort in 2015, early seedling survival did not differ between control and removal plots for plants tagged either in March (control: 70.2%, removal: 76.9%, $p = 0.59$) or in April (control: 93.9%, removal: 87.2%, $p = 0.20$). However, non-native removal significantly reduced the likelihood that seedlings had begun to drop leaves by June (control: 26.3%, removal 9.0%, $LR = 6.2$, $p = 0.013$).

Adult plants were almost exactly divided between control and removal plots from 2014–2015 (S4 Table). This ratio shifted sharply after the 2015 seedling cohort established, with 79.6% of adults found in removal plots by 2016. The same pattern held through 2020, three years after the removal treatment ended (81.9% adults in removal plots, $X^2 = 42.8$, $df = 1$, $p < 0.0001$). By 2020, mean *A. californica* canopy area per plot was nearly four times greater where non-natives had been removed from 2014–2017 ($t = 2.77$, $df = 10$, p = 0.02; control: $20.2 \pm 13.0$ m$^2$; removal: $79.7 \pm 26.6$ m$^2$).

## Discussion

Preventing type conversion after disturbance is an important management goal in southern California, particularly given substantial barriers to restoration once habitats are dominated by non-native plants [5, 21, 29, 48]. We found that non-native grass and forb removal in the first four years after a sage scrub fire facilitated recovery of the dominant native shrub by increasing survival of seedlings and small adults. We also observed strong effects of precipitation, with higher rainfall directly benefitting seedling and adult survival as well as strengthening positive responses to non-native removal by seedlings. Our results illustrate how an increasing probability of drought in the critical first few years after fire could create additional obstacles to post-fire shrub recovery in sage scrub.

Both observational and experimental studies support negative effects of non-native grasses and forbs on native seedling recruitment in sage scrub. *Artemisia californica* seedlings are largely absent from non-native annual grasslands in southern California, occurring primarily in vegetation gaps within intact sage scrub [13, 24, 25, 49]. Suppression of native seed germination by non-native grass thatch from the previous growing season is one potential cause. This effect has been shown experimentally for native forbs in competition with non-native annual grasses such as *B. diandrus* [20, 50], but not tested directly for *A. californica*. Germination of *A. californica* is inhibited in the dark, suggesting that heavy grass thatch could suppress seedling emergence [51]. Nevertheless, we did not find significant effects of non-native removal on seedling emergence. One important caveat is that our data provided low power to test for such

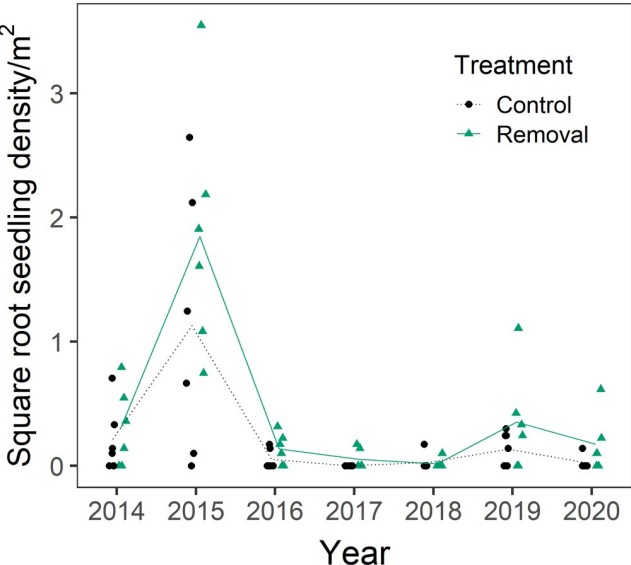

**Fig 2. Seedling densities as measured in June of each year, for control (black circles) and non-native removal (green triangles) treatments.** The lines represent annual mean values for control (black dotted) and removal (green solid). Seedling densities are shown on a square-root- transformed scale to reduce the distortion by outliers, but our statistical analysis used non-parametric permutation tests on the untransformed data. Control and non-native removal plots were sampled on equivalent dates each year; the series are offset to facilitate visual comparison.

effects because of high variation in seedling emergence among a modest number of plots and years (Fig 2). Non-native grass and forb foliar cover totaled only 17% in control plots during the first growing year after fire, so thatch levels were in any case low in fall of 2014 when most *A. californica* seedlings emerged (Table 1).

Post-germination competition with non-native annuals can also limit *A. californica* recruitment, a mechanism our results support. One previous study found that increasing grass density reduced *A. californica* seedling survival [21], while another observed complete seedling mortality unless non-native grasses were removed [52]. Our results document significant benefits of non-native removal for over-summer seedling survival. Seedlings in control plots were more likely to show evidence of water stress by onset of summer drought, consistent with previous findings that non-native grasses suppress native shrub seedlings by changing soil water

**Table 2. Results of best-fit generalized linear mixed effects models for the effects of January to June rainfall and non-native removal on adult (top) and seedling (bottom) survival.**

| Life stage | Factor | Estimate | SE | LR | *p* |
|---|---|---|---|---|---|
| Adult | Spring rainfall | 1.63 | 0.86 | 6.08 | **0.015** |
| | Exotic removal | 0.82 | 1.23 | 16.9 | **0.001** |
| | Log canopy volume | 1.04 | 0.29 | 51.6 | **0.001** |
| | Removal x Log canopy volume | -0.65 | 0.31 | 8.4 | **0.008** |
| Seedling | Spring rainfall | 0.87 | 0.9 | 18.61 | **0.001** |
| | Exotic removal | 1.78 | 0.74 | 12.51 | **0.006** |
| | Removal x rainfall | 2.08 | 1.2 | 5.62 | **0.023** |

The adult survival model included plant size, as measured by log canopy volume. Columns give the coefficient estimate and standard error determined from 1000 bootstrap replicates, as well as the likelihood ratio test statistic (LR) and p value. Bolded p values indicate results significant at a threshold of < 0.05.

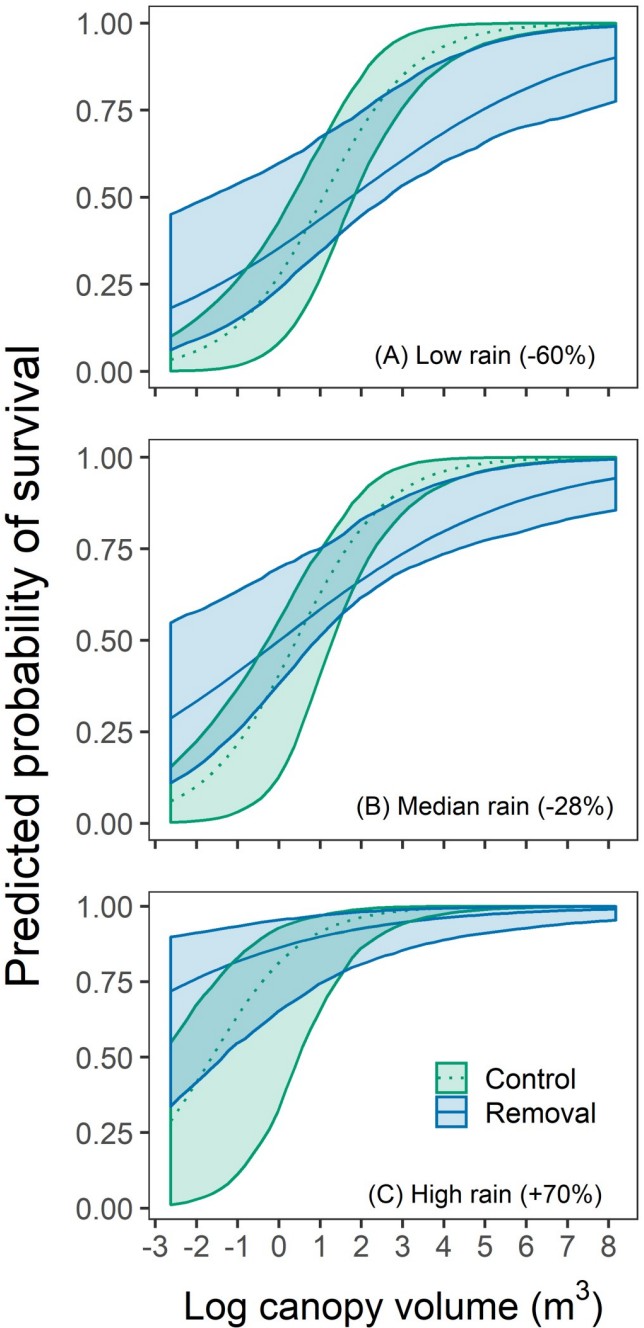

**Fig 3. Effects of non-native plant removal on the probability of survival in adults of different sizes.** The three panels show predictions for spring rainfall at (A) the study minimum (60% below the 1999–2020 average), (B) the study median (28% below average), and (C) the study maximum (70% above average). Center lines for each group show best-fit predictions, and the filled areas 95% confidence bounds based on bootstrap replicates.

availability [21]. Similarly, small adults benefitted from non-native removal, although as in other studies this effect disappeared for larger individuals [21]. At the largest sizes, survival estimates for plants in control conditions appear somewhat higher than for plants in removal conditions (Fig 3). However, many records for large plants were from individuals in removal

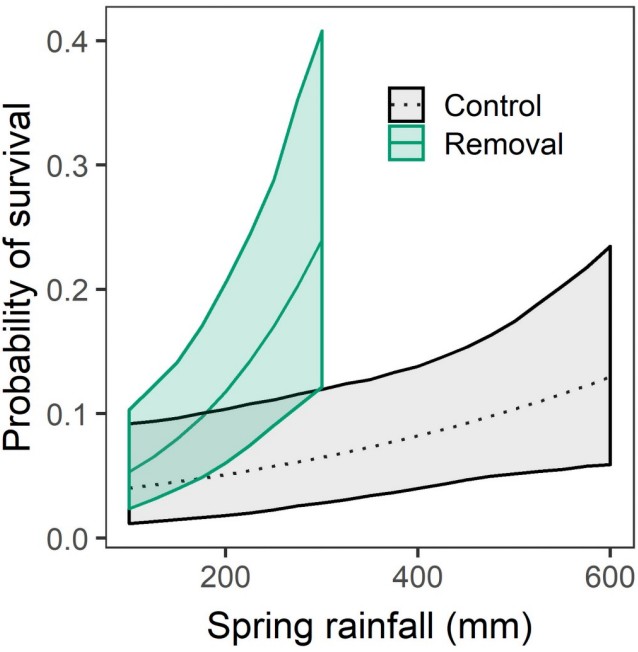

**Fig 4. Effects of spring rainfall (mm) on the probability of seedling survival, by treatment (black dashed line indicates control, green solid line non-native grass removal).** Center lines for each group represent best-fit predictions, and the filled areas 95% confidence bounds based on bootstrap replicates. Seedling survival could only be compared between control and removal treatments in years with less than 300 mm of rainfall (S1 Table). So, we did not extrapolate the model predictions for non-native removal beyond 300 mm of precipitation".

plots observed after 2017, when removal treatments stopped (70.8% of adult survival data for plants with a log canopy volume greater than 5 log [m$^3$]).

Wet years likewise increased both seedling and adult survival. Previous studies similarly document the importance of water limitation to native shrub communities. Drought years have been associated with reduced post-fire survival of resprouts in chaparral [30] and low native shrub seedling recruitment in sage scrub [24]. Experimentally lowering water availability depressed growth rates and increased allocation to below-ground biomass in *A. californica* seedlings [53]. Detrimental effects of water stress on adult shrubs in chaparral and sage scrub can reduce canopy cover and increase vulnerability to invasion [26, 54], as well as slow post-fire regeneration of cover [34]. Still, no previous study has to our knowledge quantified demographic effects of drought in southern California across shrub life history stages, a critical step for modeling post-fire recovery. Two patterns from our data seem particularly noteworthy. First, rainfall effects on seedling survival interacted strongly with non-native removal, suggesting little benefit of wet years for recruitment unless non-native competitors are controlled. Second, larger plants survived better than smaller, more recently established ones but still experienced substantial mortality in dry years (Table 2, Fig 1).

The balance between resprouting and recruitment from seed plays a critical role in shaping post-fire recovery of chaparral and sage scrub. Sage scrub is considered resilient and can quickly recover to pre-fire composition, so long as enough individuals crown resprout [5]. Yet resprouting rates vary dramatically across habitats and individual shrub species [43]. We observed a surprisingly small number of resprouts given the high pre-fire cover of *A. californica* (*N* = 7 across 0.12 hectares). While *A. californica* resprouts after fire less than other sage scrub shrubs such as *Salvia apiana* and *Eriodictyon trichocalyx* var. *trichocalyx*, rates

documented in other studies still range from 13% to 25% [42, 43]. Using size to classify resprouts rather than excavating around roots may have led to underestimation of resprouts in our data. Still, individuals present in the first year after fire represented only a small fraction of those remaining in 2020 (12.4%). This 2013–2014 cohort experienced poor survival and contributed little to *A. californica* cover in our plots.

Several factors could explain low resprouting rates at this site. First, shrubs may resprout less at inland locations compared to coastal ones, potentially because of past selection due to lower historical fire frequencies in drier habitats [55]. This pattern is not consistently supported for *A. californica* by previous studies, however [42]. Stand age is another potential explanation, as older individuals likely lose their capacity to resprout [33, 43]. We do not know the age distribution of *A. californica* before the 2013 fire, but no other major disturbance events likely to generate stand replacement had occurred in at least 60 years. Previous work argues that most populations of *A. californica* are relatively even-aged, although assigning ages to individual plants is complicated by their tendency to resprout even in the absence of fire [56]. Regardless of the cause, our data reinforce that crown resprouting is highly variable across sites.

When resprouting rates are low, native shrub recovery hinges on recruitment from seed. This may make communities more vulnerable to increased fire frequency if time between fires is insufficient for seedbanks to replenish [48]. Since seedlings are particularly vulnerable to effects of non-native species, experience high mortality (Fig 3), and contribute less cover, communities with low resprouting rates are likely more vulnerable to type conversion after fire [57]. Pulse seedling recruitment is common for obligate seeding shrubs in both chaparral and sage scrub, typically in the first year after fire [42]. Strong reproduction by resprouting shrubs immediately after fire can also lead to a large peak in germination during the second year [41, 58]. Given limited resprouting, high seedling emergence in the second post-fire year at our site likely came from the seed bank. Dispersal beyond the vicinity of a parent shrub is thought unusual in most sage scrub species [5]. In contrast to well-studied, larger-scale fires, unburned habitat remained within a few hundred meters of all our plots (Fig 1). Still, the large drop in seedling emergence after 2014–2015 suggests depletion of the seed source; 90% of the seedling density in our plots concentrated into this single growing year. Keeley [40] found that more than 80% of *A. californica* seedlings recorded within 5 years after fire emerged in the first two years.

Delay of seedling emergence into the second year at our site may have been caused by low rainfall immediately after the fire occurred, in the window between November and January when most germination takes place. Precipitation during these months in 2013–2014 was 72% below the mean for 1999–2020 (2013–2014: 58.41 mm; 1999–2020 mean: 207.2 ± 38.9 mm). During the following growing year when *A. californica* emerged in large numbers, early rainfall increased by more than three times (177.5 mm). In general, early rain likely to stimulate germination (November through January) correlates with January to June precipitation linked to seedling and adult survival (1999–2020, $r = 0.75$, $df = 20$, $p < 0.0001$). This creates potential for germination cuing to reduce the risk of emergence into a low survival year. Still, our data show that early precipitation is not always a reliable cue of spring conditions. The small number of seedlings germinating in 2013–2014 benefitted from much higher January to June rainfall (277.8 mm; S3 Table) than the large number of seedlings germinating in 2014–2015 (114.7 mm).

In summary, our findings reinforce the high vulnerability of sage scrub to post-fire loss of shrub cover and potential type conversion, particularly with increasing drought frequency and in stands with low rates of crown resprouting. Yet they also illustrate the potential for targeted management of non-natives immediately after fire to promote recovery of native shrubs.

Caution in extrapolating these results is important, given the small scale of both the study area and fire. At the same time, most remaining intact sage scrub habitat in southern California consists of small fragments bordering on urban and suburban development [39, 40]. Our work helps fill spatial gaps in previous research, given other studies have mostly concentrated in coastal areas or further east near the boundary of sage scrub distributions. Additional studies in similar small fragments will help improve our understanding of the diverse successional pathways in these threatened shrub communities.

## Supporting information

**S1 Table. Survival of seedlings from when they were marked in June of their first year to the following June.** Columns show the year of germination (spring); total spring rainfall (January to June) for that year (mm); treatment (control or non-native removal); the number of marked seedlings that survived; the number of marked seedlings that died; and the percent survival. Percent survival values were only calculated for years in which at least 20 seedlings were tagged for each treatment. Non-native removal treatments were applied in 2014–2017 but not 2018–2019. For 2018–2019, we denoted treatment as "None", with the original treatment assignment in parentheses (C = control, R = removal).
(DOCX)

**S2 Table. Summary of all mixed effects survivorship models tested, for both seedlings and adults.** The model specifications are given, with fixed effect predictors abbreviated as Size (log transformed canopy volume in $m^3$), Treat (removal or control treatment), and Rain (rainfall from January through June). Random effects for Plot were included in all models, and random effects for Tag (individual plant identity) in the adult models. Terms that changed from the best-fit model are shown (- for removed, + for added), along with the AIC values.
(DOCX)

**S3 Table. Percent cover of the most common non-native species in control and removal plots for the first four spring surveys (late March to April) after the October 2013 fire.** The most common non-native forb *Erodium* spp. was unaffected by the removal treatment and is not included; cover values for *Erodium* can be found in Table 1. Species are annotated by life history (G = non-native annual grasses, F = non-native annual forbs, SS = non-native sub-shrubs).
(DOCX)

**S4 Table. Distribution of established (greater than one year old)** *Artemisia californica* **between control and removal plots over the 7 post-fire study years.** Numbers for each treatment represent the total plants across all plots in that treatment (*N* = 6). The percent of plants in removal plots is also shown for each year.
(DOCX)

## Acknowledgments

We thank Nancy V. Hamlett for assistance with project implementation and plant identification, as well as volunteers at the Bernard Biological Field Station for their work applying the non-native plant removal treatments. Lauren Hartz, Emily Audet, Nico Tutland, Wendy Norena, Mary-Clare Bosco, and a number of other undergraduate students at the Claremont Colleges contributed to collection of plant cover data.

## Author Contributions

**Conceptualization:** Diane M. Thomson, Wallace M. Meyer, III.

**Data curation:** Diane M. Thomson.

**Formal analysis:** Diane M. Thomson.

**Investigation:** Diane M. Thomson, Wallace M. Meyer, III, Isobel F. Whitcomb.

**Methodology:** Diane M. Thomson.

**Visualization:** Diane M. Thomson.

**Writing – original draft:** Diane M. Thomson.

**Writing – review & editing:** Wallace M. Meyer, III.

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
