## [Decision Letter · Decision Letter 0]

27 Apr 2021

PONE-D-21-06311

Non-native plant removal and high rainfall years promote post-fire recovery of *Artemisia californica*in southern California sage scrub

PLOS ONE

Dear Dr. Thomson,

Thank you for submitting your manuscript to PLOS ONE. After careful consideration, we feel that it has merit but does not fully meet PLOS ONE’s publication criteria as it currently stands. Therefore, we invite you to submit a revised version of the manuscript that addresses the points raised during the review process (see below).

We look forward to receiving your revised manuscript.

Kind regards,

Cristina Armas

Academic Editor

PLOS ONE

Journal Requirements:

We note that you have stated that you will provide repository information for your data at acceptance. Should your manuscript be accepted for publication, we will hold it until you provide the relevant accession numbers or DOIs necessary to access your data. If you wish to make changes to your Data Availability statement, please describe these changes in your cover letter and we will update your Data Availability statement to reflect the information you provide.

We note that Figure 1 in your submission contain satellite images which may be copyrighted. All PLOS content is published under the Creative Commons Attribution License (CC BY 4.0), which means that the manuscript, images, and Supporting Information files will be freely available online, and any third party is permitted to access, download, copy, distribute, and use these materials in any way, even commercially, with proper attribution. For these reasons, we cannot publish previously copyrighted maps or satellite images created using proprietary data, such as Google software (Google Maps, Street View, and Earth). For more information, see our copyright guidelines: http://journals.plos.org/plosone/s/licenses-and-copyright.

Additional Editor Comments :

Comments besides those from reviewers:

Please add line numbers in your next manuscript version

Page 4. “non-native effects on fire”. Too vague. Something is missing here. On what fire characteristic do non-native plant species had an effect?

“Once type conversion occurs, subsequently restoring shrubs through management has proved very difficult” Please simplify this sentence. Moreover, “type conversion” or “type-converted” are terms/concepts/processes used in many instances in the introduction and in the concluding paragraph in the discussion. Thus, please be precise and define it.

I agree with one of the reviewers, the International System of Units (SI) suggests using mm instead of cm for precipitation

Page 17. Please, rephrase this sentence in a more formal tone. There are no control or removal plants. “At the largest sizes, survival estimates for control plants appear somewhat higher than for removal plants (Fig 3).” The same occurred for the next sentence in that paragraph.

Figures

Fig. 2 displays the square-root transformed densities of seedlings across years and exotic-removal treatments. What is the purpose of this transformation of the data? In page 11 (data analysis) authors state that transformation of the data did not normalize the distribution of errors.

Legend of Fig. 4. Please rephrase the last sentence, unclear “Removal treatments were not observed under conditions of greater than 300 mm spring rainfall, so model results are not extrapolated to higher values.”

Reviewers' comments:

Reviewer's Responses to Questions

**Comments to the Author**

1. Is the manuscript technically sound, and do the data support the conclusions?

Reviewer #1: Yes

Reviewer #2: Partly

2. Has the statistical analysis been performed appropriately and rigorously? 

Reviewer #1: Yes

Reviewer #2: Yes

3. Have the authors made all data underlying the findings in their manuscript fully available?

Reviewer #1: Yes

Reviewer #2: Yes

4. Is the manuscript presented in an intelligible fashion and written in standard English?

Reviewer #1: Yes

Reviewer #2: Yes

5. Review Comments to the Author

Reviewer #1: In this manuscript, entitled “Non-native plant removal and high rainfall years promote post-fire recovery of Artemisia californica in southern California sage scrub”, authors conducted a long term experimental field study to evaluate the post-fire recovery of the shrub A. californica in a Mediterranean plant community in California, by assessing the interactive effects of non-native plants removal and wet/dry years on A. californica seedling emergence, recruitment, and adult plants survival and growth. The manuscript is well written. The experiment is well designed and the results are sound based on data analysis and support the conclusions. Overall, the manuscript introduces novel evidence that underlines adequate practices (i.e., invasive species removal) in post-fire management to avoid excessive habitat loss by non-native plant invasion under foreseen climatic drought scenarios. Comments below may help to improve the manuscript:

In “study system” section, authors provide data on annual rainfall (September to August) in cm (i.e., 41.55 cm per year), while a couple of sentences below they provide data in mm (i.e., 41.8 mm). I guess there is a typo somewhere. Afterwards, they refer to table S1, where they provide data on Spring rainfall as mm. However, data scales to hundreds of mm. Please, revise the data and use always the same units (preferably in mm). Also, please, correct the typo with data period in …complete records; 1986-197? NOOA…

I find that Fig 1 would be more informative by directly including plots and treatments distribution.

In table 2, does “exotic removal” refers to “non-native removal” treatment? Please clarify terminology.

Reviewer #2: The manuscript ref PONE-D-21-06311 describes the monitoring of post-fire seedling emergence and survival as affected by non-native grasses and forbs removal during a 7-yr period. It relates the probability of survival of both seedlings and adults of Artemisia californica with spring rainfall and plant size in a yearly basis, with interesting results. The main problem I see is the lack of replication as the whole study is conducted in 12 plots at the same location which reduces the transferability of results. In fact, authors state that “Seedling emergence varied strongly spatially” (Page 14) and “Caution in extrapolating these results is important, given the small scale of both the study area and fire” (Page 20). However, the large monitoring period provides extra value to the study.

Introduction section is well-structured and clearly establishes the objectives and hypotheses.

- It is not clear to me whether the ratio resprouting/recruitment of A.californica is higher inland or in coastal areas as I do not know how is the moisture gradient (I can guess but better if it is explicitly stated). Page 6

- Table S1, what does ‘none’ mean in the treatment column for years 2018 and 2019? It appears in Page 11 but should be define in the table heading (or in a footnote)

- Did you monitor the biomass of the non-natives removed? This could be used as covariable.

- Were those two overdense plots that precluded tagging all individuals paired? Page 9

- "The two westernmost plots (one control, one removal) were surveyed between February 20 and March 1, and the next pair of plots between March 12 and April 3. A second control plot was added to the early April tagging, to increase sample sizes for control seedlings". Recruitment in the latest sampled control plot was ca. 5 weeks after the earliest which can result in overestimation.

- "but resulting drops in sample size and the loss of observations from the very wet spring of 2019 would have reduced statistical significance". This is speculative

- Not sure canopy volume can be considered fixed factor

Table 1. Remove SE columns and include the numbers in brackets (or after ±) next to the mean value. Include different letters for significant differences between control and removal treatments

- "Seven A. californica were confirmed as resprouts in June 2014, 3 in control and 4 in removal plots. Almost all individuals established after the fire from seedlings". What was the percentage of germinated seedlings? These numbers are more precise than ‘almost all’.

Figure 2. It seems that density of recruits in removal plots were recorded later than in control plots. Correct it

- "absence of non-native grass removal", in page 17, sounds weird these two (three in fact) negations in the sentence

6. PLOS authors have the option to publish the peer review history of their article (what does this mean?). If published, this will include your full peer review and any attached files.

Reviewer #1: No

Reviewer #2: No

---

## [Author Response · Author response to Decision Letter 0]

10 Jun 2021

Dear Dr. Armas,

I am writing to submit a revised version of the MS, “Non-native plant removal and high rainfall years promote post-fire recovery of Artemisia californica in southern California sage scrub”. My co-authors and I greatly appreciated the thoughtful comments from you and the reviewers. I believe the revisions address all of the questions and recommendations, but if the changes did not fully hit the mark in some respects please let me know.

I have condensed the original text of the reviews in places to facilitate reading; paragraphs of text with our responses are all denoted with a – mark.

Thank you again for your time helping us with this MS,

Diane Thomson

Journal Requirements:

- We rechecked that style requirements were met and now use the correct format for file names as well as Figure and Table legends. We also removed the Author Contributions section from the MS body, as it seemed this was not meant to be included there. Please advise if we have misinterpreted any details of the instructions; this is our first submission to PLOS, and we are still learning the formatting.

- We are working on the Dryad repository for this data set, and will complete it immediately if the MS is accepted for publication.

3. We note that Figure 1 in your submission contain satellite images which may be copyrighted. 

- We have removed the images and remade Figure 1.

4. Please review your reference list to ensure that it is complete and correct. 

- We checked the references carefully. We found and corrected some minor typos and errors in the references, but made no major change. Note that we corrected errors in the references after accepting the other changes to the MS, so they do not appear in the Track Changes version of the revision.

Additional Editor Comments:

Please add line numbers in your next manuscript version

- We added continuous line numbers.

Page 4. “non-native effects on fire”. Too vague. Something is missing here. On what fire characteristic do non-native plant species had an effect?

- We have revised this sentence to connect more explicitly with the section on page 2 where potential effects of non-native plants on fire are discussed in more detail (revision page 4, lines 66-70).

“Increased fire frequency is associated with higher non-native grass and forb cover in both chaparral and sage scrub [14,16–18]. Yet, the directions of any causal relationships remain unclear; these correlations could reflect the effects of fire disturbance on non-native plants, the effects of non-native plants on ignition risk and fuel loads, or both.”

“Once type conversion occurs, subsequently restoring shrubs through management has proved very difficult” Please simplify this sentence. Moreover, “type conversion” or “type-converted” are terms/concepts/processes used in many instances in the introduction and in the concluding paragraph in the discussion. Thus, please be precise and define it.

- We simplified the sentence as requested. Page 4, lines 78-79.

“Reintroducing shrubs to restore type-converted, non-native grasslands has proved very difficult [5,29].”

- Yes, the term type conversion is introduced early on in the introduction (the second paragraph) and then used regularly. We expanded on the definition given with this first use of the term. Page 2, lines 56-59.

“These changes in fire regime can lead to habitat degradation and even type conversion, meaning a major and persistent shift in community structure such as replacement of native shrubland by non-native grassland [12].”

I agree with one of the reviewers, the International System of Units (SI) suggests using mm instead of cm for precipitation

- We changed all precipitation measures to mm, as requested.

Page 17. Please, rephrase this sentence in a more formal tone. There are no control or removal plants. “At the largest sizes, survival estimates for control plants appear somewhat higher than for removal plants (Fig 3).” The same occurred for the next sentence in that paragraph.

- We rephrased as follows (revised MS, page 17 lines 369-373). Please advise if this wording does not fully address the comment.

“At the largest sizes, survival estimates for plants in control conditions appear somewhat higher than for plants in removal conditions (Fig 3). However, many records for large plants were from individuals in removal plots observed after 2017, when removal treatments stopped (70.8% of adult survival data for plants with a log canopy volume greater than 5 log [m3]).”

Figures

Fig. 2 displays the square-root transformed densities of seedlings across years and exotic-removal treatments. What is the purpose of this transformation of the data? In page 11 (data analysis) authors state that transformation of the data did not normalize the distribution of errors.

- The purpose of the transformation is to facilitate interpretation of the data. In a plot of the untransformed data, the tail of unusual high values pulls the y axis out at the top and leaves all the other points concentrated near the bottom. The permutation methods we used for the formal statistical analysis have effects similar to transformation, in that they reduce the influence of unusual values. In that sense, the plot of transformed data is a better way of visualizing the analysis we performed than a plot of the untransformed data would be. We have added a sentence to the legend explaining why the data are presented on a square-root- transformed scale, even though the statistical analysis used the untransformed data (revised MS page 14, lines 296-298).

“Seedling densities are shown on a square-root-transformed scale to reduce the distortion by outliers, but our statistical analysis used non-parametric permutation tests on the untransformed data.”

Legend of Fig. 4. Please rephrase the last sentence, unclear “Removal treatments were not observed under conditions of greater than 300 mm spring rainfall, so model results are not extrapolated to higher values.”

- We revised this part of the legend. Please advise if the new wording is not clear enough. Revised MS page 15, lines 326-328.

“Seedling survival could only be compared between control and removal treatments in years with less than 300 mm of rainfall (Table S1). So, we did not extrapolate the model predictions for non-native removal beyond 300 mm of precipitation.”

Reviewer #1: 

In “study system” section, authors provide data on annual rainfall (September to August) in cm (i.e., 41.55 cm per year), while a couple of sentences below they provide data in mm (i.e., 41.8 mm). I guess there is a typo somewhere. Afterwards, they refer to table S1, where they provide data on Spring rainfall as mm. However, data scales to hundreds of mm. Please, revise the data and use always the same units (preferably in mm). Also, please, correct the typo with data period in …complete records; 1986-197? NOOA…

- Thanks for catching the error; the original MS intended to report both the precipitation values cited above in cm. We have changed all precipitation measures to units of mm. The typo in the year range for complete rainfall records has also been corrected.

I find that Fig 1 would be more informative by directly including plots and treatments distribution.

- The figure has been modified (also requested by the editor) and now shows individual plots and treatments.

In table 2, does “exotic removal” refers to “non-native removal” treatment? Please clarify terminology.

- Yes. We changed Table 2 so it consistently uses “non-native removal” throughout.

Reviewer #2: The main problem I see is the lack of replication as the whole study is conducted in 12 plots at the same location which reduces the transferability of results. In fact, authors state that “Seedling emergence varied strongly spatially” (Page 14) and “Caution in extrapolating these results is important, given the small scale of both the study area and fire” (Page 20). However, the large monitoring period provides extra value to the study.

- We agree that our study is local in scale; that is why we included the caveats cited above. But many if not most ecological studies are limited to a single site, particularly research on uncontrolled disturbance events like fire. Almost all the experimental studies and many of the post-fire studies this MS cites are from a single site. Both the level of replication (plots/treatment) and the spatial dimensions of sampling in our work are just as or even more extensive than in these comparable studies (detailed examples can be found at the end of this response letter). Even the results of multi-site research are not necessarily broadly transferable. Arguably, one of the best ways to build a generalized understanding is through synthesizing many local studies like this one. We have added another sentence noting the value of our work in filling geographic gaps relative to the distribution of previous studies across sage scrub habitats in southern California. (Revised MS page 20 lines 443-445).

“Our work helps fill spatial gaps in previous research, given other studies have mostly concentrated in coastal areas or further east near the boundary of sage scrub distributions.”

It is not clear to me whether the ratio resprouting/recruitment of A.californica is higher inland or in coastal areas as I do not know how is the moisture gradient (I can guess but better if it is explicitly stated). Page 6

- We added some language to clarify this point. Revised MS, page 6 lines 123-125.

“Several studies suggest that the ratio of resprouting to recruitment from seed in sage scrub declines across a moisture gradient, from more mesic coastal to drier inland habitat [33,43].”

Table S1, what does ‘none’ mean in the treatment column for years 2018 and 2019? It appears in Page 11 but should be define in the table heading (or in a footnote)

- The non-native removal treatment was applied in 2014-2017, but not during the final two years of data collection in 2018-2019. We added an explanation to the S1 Table legend. (MS lines 456-458).

“Non-native removal treatments were applied in 2014-2017 but not 2018-2019. For 2018-2019, we denoted treatment as “None”, with the original treatment assignment in parentheses (C=control, R=removal).”

Did you monitor the biomass of the non-natives removed? This could be used as covariable.

- No, we did not measure the biomass of the non-natives removed; this would have been time consuming given the size of our plots (each 100 m2). Our analyses show clear differences in seedling and adult survival between control and removal plots. Adding covariates might have accounted for some additional background variation/noise, but it seems to us very unlikely this would change our core findings.

- Were those two overdense plots that precluded tagging all individuals paired? 

- Yes. We clarified this point in the text. Page 9 lines 188-190.

“For one control and one removal plot adjacent to each other (6 and 7)…”

"The two westernmost plots (one control, one removal) were surveyed between February 20 and March 1, and the next pair of plots between March 12 and April 3. A second control plot was added to the early April tagging, to increase sample sizes for control seedlings". Recruitment in the latest sampled control -plot was ca. 5 weeks after the earliest which can result in overestimation.

- We want to make sure there is no ambiguity about when the recruitment data (seedling densities, as reported in Fig. 2) were collected: at the same time for all plots and years, during the same June census used for adult plants (page 8, lines 188-189). The passage quoted above refers to a different component of our work. We marked a subset of seedlings present in March and April of 2015 to test for treatment effects on early-season survival (between emergence and the June census). To better clarify this distinction, we added a sentence to the section on seedling data collection (page 10, lines 207-208). We also separated the information on early-season seedling tagging in 2015 into a different paragraph, starting on line 209.

“All seedling density data were collected in the same June census used for adult plants.”

- Returning to the effects of tagging seedlings in two different months: yes, the observed survival was lower for plants we tagged earlier. That is why we reported the results separately for seedlings tagged in March and April (see page 14, lines 305-308, copied in below). The key point is that the control and removal seedlings whose survival we compared were tagged at the same time, for both the March and April cohorts.

“For the largest cohort in 2015, early seedling survival did not differ between control and removal plots for plants tagged either in March (control: 70.2%, removal: 76.9%, p = 0.59) or in April (control: 93.9%, removal: 87.2%, p = 0.20).”

- "but resulting drops in sample size and the loss of observations from the very wet spring of 2019 would have reduced statistical significance". This is speculative.

- We should have been more explicit in the original MS and have revised to clarify. We re-ran both the adult and seedling survival analyses without the 2018-2019 data, to make sure that including those years (after the experimental manipulations had ended) did not drive our findings. The p values do change slightly, of course, but none of the main results are different. The key point is that our findings are robust to classifying the 2018-2019 records as “control” (no non-native removal) data points. We simplified the text to say that and nothing else. Revised MS, page 11, lines 236-239.

- “To make sure this decision did not drive any of our findings, we also ran the adult and seedling survival analyses with data only from the time period when experimental manipulations were carried out (2014-2017). None of the qualitative results for effects of precipitation, non-native removal, and plant size change when data from 2018-2019 are excluded.” 

Not sure canopy volume can be considered fixed factor

- We were not quite sure how to interpret this comment; is the idea that canopy volume instead should be a random effect? We do not think it is possible to treat canopy volume (plant size) as a random effect in our models. First, random effects vary across individuals, while fixed effects do not. Treating canopy volume as a random factor would mean the effects of size on survival vary for each individual plant. On the other hand, block or plot identities are regularly treated as random effects. For example, in our analyses each plot (random effect) has a different intercept. Further, random effects in mixed models generate estimates of variance for the estimated distribution of intercepts/slopes, but not specific coefficients. One common way to describe fixed effects is as predictors researchers are “interested in”, meaning they aim to estimate coefficients and test hypotheses for those variables. For example, we could not have made Figures 3 and 4 without treating canopy volume as a fixed effect, one that has a specific coefficient (slope of the linear relationship with a logit link, in this case).

- The approach we used is fairly common in demographic modeling for plants. Below we give a small sampling of references from the many studies where authors likewise fit GLMM statistical models for survival and growth, with plant size and sometimes climate variables as fixed effects and individual plot or year as random effects.

- Dalgleish et al. 2011. Climate influences the demography of three dominant sagebrush steppe plants. Ecology 92: 75-84.

- Miller et al. 2012. Evolutionary demography of iteroparous plants: incorporating non-lethal costs of reproduction into integral projection models. Proc. R. Soc. B. 2792831–2840.

- Mandel and Ticktin 2012. Interactions among fire, grazing, harvest and abiotic conditions shape palm demographic responses to disturbance. Journal of Ecology 100 (997-1008).

Table 1. Remove SE columns and include the numbers in brackets (or after ±) next to the mean value. Include different letters for significant differences between control and removal treatments.

- We changed the columns in Table 1 as requested. It is not possible to put letters indicating significant differences, because the statistical analysis we used to evaluate treatment effects on cover combined all years, while Table 1 shows each year separately. We think Table 1 is a helpful supplement to the statistical results reported in the main text; it shows the changes in both control and non-native removal plots over time, and the non-significant trend towards a time by treatment effect. 

"Seven A. californica were confirmed as resprouts in June 2014, 3 in control and 4 in removal plots. Almost all individuals established after the fire from seedlings". What was the percentage of germinated seedlings? These numbers are more precise than ‘almost all’.

- We revised as follows (page 14, lines 286-288):

“Seven A. californica were confirmed as resprouts in June 2014, 3 in control and 4 in removal plots. The other 454 adult plants identified and tagged from 2014-2019 recruited from seed (98.5 % of total).”

Figure 2. It seems that density of recruits in removal plots were recorded later than in control plots. Correct it.

- We offset the points and lines for the two treatments in this plot because otherwise they crowd each other out, making the figure difficult to read. (Because many of the values overlap near 0, we also needed to jitter the individual data points). We have revised the figure to reduce the amount of offset (distance between the two data series), and added a sentence to the legend clarifying that the sampling dates were identical for the two treatments. Note that offsetting in the R ggplot package always puts the series equidistant on either side from the x axis marker. Please advise if these changes do not fully address the concern. Page 14, lines 298-299.

“Control and non-native removal plots were sampled on equivalent dates each year; the series are offset to facilitate visual comparison.”

- "absence of non-native grass removal", in page 17, sounds weird these two (three in fact) negations in the sentence

- We rephrased the sentence (Page 17, lines 362-364).

“One previous study found that increasing grass density reduced A. californica seedling survival [21], while another observed complete seedling mortality unless non-native grasses were removed [52].”

Examples of replication and sampling area from similar research

Our study: 1 site. Experimental and post-fire shrub monitoring. Study area ~ 300 m by 10 m. 6 plots and 600 m2 sampled per treatment. 1006 seedlings and 461 adult plants tagged (7 years).

Conlisk 2016 (PLoS One): 1 site. Post-fire monitoring. Study area ~ 200 by ~120 m. Four transects in burned area, four in unburned, each 100 m in length with 5 quadrats. 38 m2 sampled per ”treatment”. 

Molinari and D’Antonio (Biol Invasions) 2019. 1 site. Experimental. Study area in 8 blocks at least 5 m apart, each 3 m2 . 8 plots and 1.28 m2 sampled per treatment.

Minnich 2014 (Global Change Biology). 1 site. Post-fire monitoring. Study area not specified. 416 plants tagged across all shrub species, max 200 per species (6 years).

Eliason and Allen 2008 (Restoration Ecology): 1 site. Experimental. Total study area 12 m by 13 m. 3 m2 sampled per treatment.

Cox and Allen 2008 (Journal of Applied Ecology): 1 site. Experimental. Study area ~2 hectares. 5 plots and 500 m2 sampled per treatment.

DeSimone and Zedler, 1999 (Ecology): 1 site. Experimental. Study area not specified. 8 plots and 8 m2 sampled per treatment.

Keeley and Keeley 1984 (American Midland Naturalist). 2 sites, both in Santa Monica Mountains, southern California. Postfire shrub survey., 10 plots and 80 m2 sampled. Total number of plants measured at all sites/for all species=857 (single time point).

Malanson and O’Leary 1982 (Oecologia): 6 sites, all at Santa Monica Mountains, southern California. Postfire monitoring. Each site 24 m2 in area. Sampled 16 quadrats and 64 m2 per site.

---

## [Editor Report · Decision Letter 1]

28 Jun 2021

Non-native plant removal and high rainfall years promote post-fire recovery of *Artemisia californica*in southern California sage scrub

PONE-D-21-06311R1

Dear Dr. Thomson,

We’re pleased to inform you that your manuscript has been judged scientifically suitable for publication and will be formally accepted for publication once it meets all outstanding technical requirements.

Kind regards,

Cristina Armas

Academic Editor

PLOS ONE
---

## [Editor Report · Acceptance letter]

13 Jul 2021

PONE-D-21-06311R1 

Non-native plant removal and high rainfall years promote post-fire recovery of *Artemisia californica* in southern California sage scrub 

Dear Dr. Thomson:

I'm pleased to inform you that your manuscript has been deemed suitable for publication in PLOS ONE. Congratulations! Your manuscript is now with our production department. 

Kind regards, 

on behalf of

Dr. Cristina Armas 

Academic Editor

PLOS ONE